# Biopreservation of Fresh Sardines (*Sardina pilchardus*) Using *Lactiplantibacillus plantarum* OV50 Isolated from Traditional Algerian Green Olives Preparations

**DOI:** 10.3390/foods13030368

**Published:** 2024-01-23

**Authors:** Nassima Mohellebi, Samia Hamma-Faradji, Kamel Bendjeddou, Amel Ait Meddour, Yassine Benchikh, Farida Bendali, Yanath Belguesmia, Djamel Drider

**Affiliations:** 1Université de Bejaia, Faculté des Sciences de la Nature et de la Vie, Laboratoire de Microbiologie Appliquée, 06000 Bejaia, Algeria; nassima.mohellebi@univ-bejaia.dz (N.M.); kamel.bendjeddou@univ-bejaia.dz (K.B.); ait_meddour_amel@yahoo.fr (A.A.M.); farida.bendali@univ-bejaia.dz (F.B.); 2Université de Bejaia, Faculté des Sciences de la Nature et de la Vie, Laboratoire de Biochimie Appliquée, 06000 Bejaia, Algeria; yassine.benchikh@umc.edu.dz; 3Laboratoire de Biotechnologie et Qualité des Aliments, Institut de la Nutrition, de l’Alimentation et des Technologies Agro-Alimentaires (INATAA), Université Constantine 1 Frères Mentouri, 25000 Constantine, Algeria; 4Unité Mixte de Recherche (UMR) Transfrontalière BioEcoAgro1158, Univ. Lille, INRAE, Univ. Liège, UPJV, YNCREA, Univ. Artois, Univ. Littoral Côte D’Opale, ICV—Institut Charles Viollette, 59000 Lille, France; yanath.belguesmia@univ-lille.fr

**Keywords:** lactic acid bacteria, *Lactiplantibacillus plantarum* OV50, biopreservation, coculture, sardines

## Abstract

*Lactiplantibacillus plantarum* OV50 is a novel strain that was isolated from Algerian olives. Prior to its use as a natural biopreservative, OV50 underwent characterization for various functions. OV50 shows no proteolytic, lipolytic, or hemolytic activity. In addition, it is non-cytotoxic to eukaryotic cells and does not exhibit acquired antibiotic resistance. OV50 was tested with *Pseudomonas aeruginosa* ATCC 27835, *Staphylococcus aureus* ATCC 6538, *Escherichia coli* ATCC 8739, and *Vibrio cholerae* ATCC 14035 in a sardine based-medium at 37 °C and 7 °C. At 37 °C, OV50 completely inhibited the growth of these foodborne pathogens for a maximum of 6 h. At 7 °C, it suppressed their growth for a maximum of 8 days, except for *S. aureus* ATCC 6538, whose growth was reduced from 4 to 2 log CFU/mL. Microbiological counts, total volatile basic nitrogen (TVB-N), and peroxide values (PV) concentrations were determined in fresh sardines inoculated with OV50 and kept at 7 °C for 12 days. The inoculated sardines showed a significant reduction in TVB-N levels at D8 (34.9 mg/100 g) compared to the control (59.73 mg/100 g) and in PV concentrations at D4 (6.67 meq/kg) compared to the control (11.44 meq/kg), as well as a significant reduction in the numbers of *Enterobacterales*, Coliforms, *Pseudomonas* spp., *Vibrio* spp., and *S. aureus* At D8 and D12 compared to the control. Taken together, these results indicate that OV50 can improve the microbiological safety, freshness, and quality of sardines.

## 1. Introduction

Sardina pilchardus is a sardine found in the Mediterranean Sea and is widely traded and consumed in the region [1]. This small pelagic fish is rich in proteins, minerals, and polyunsaturated fatty acids, especially Ω-3 [2,3]. Sardines have a short and limited shelf life due to their delicate structure, neutral pH, and chemical composition [2]. As a result, fresh sardines rapidly perish even under refrigerated storage conditions. Their quality may deteriorate rapidly due to microbial activity, autolysis, and oxidation occurring during the storage conditions [2,4]. In addition, post-harvest activities within the fish chain have often been neglected in rural community development projects. However, fish deserves more attention than it currently receives [5]. Post-harvest fish losses (PHFL) are a major challenge in the seafood industry. In fact, PHFL is a decrease in the quantity, quality, or monetary value of fish that occurs from the time of capture to the final consumer, and this phenomenon poses a significant socio-economic challenge that not only represents a loss of potential revenue for industry participants but also has adverse consequences for food security [6]. Relatedly, post-harvest spoilage is estimated to be 10–20 million tons of fish discarded annually worldwide, and losses in developing countries can be as high as 50% of domestic fish production [7]. To reduce these losses, fish can be preserved using several techniques, including low temperature, chemical treatment, and irradiation [8]. Seafood is highly perishable, even under refrigerated storage conditions. The irradiation process is expensive and can induce changes that may alter the chemical composition and nutritional value of foods [9]. With the banning of antibiotics and chemical preservatives, coupled with consumer preferences for naturalness, environmental concerns, and the desire for fresh foods, there has been a significant shift in research focus over the past decade. Researchers are now exploring natural preservatives for seafood [9,10]. Lactic acid bacteria (LAB) are good candidates for use as natural preservatives to provide a more sustainable, health-conscious, and environmentally friendly alternative to synthetic preservatives. In fact, these natural biopreservatives are known to produce several metabolites of interest, including bacteriocins, diacetyl, reutericyclin, organic acids, acetoin, and hydrogen peroxide, which are endowed with inhibitory properties [11], that enable them to reduce enzymatic and oxidative spoilage and inhibit the growth of some foodborne pathogens and spoilage microorganisms [12]. Bacteriocins act as natural antibiotics, creating a protective barrier against potential spoilage microorganisms. On the other hand, organic acids, particularly lactic acid, contribute to lowering the pH of the environment, creating conditions that are unfavorable for the survival and proliferation of spoilage microorganisms and pathogens. The lowered pH disrupts the cellular functions of these unwanted microbes, preventing their growth and product spoilage [11,12].

LAB are good candidates for use as natural preservatives in seafood. In addition, many LAB strains that are considered safe for consumption are able to grow at refrigerated temperatures. They have the ability to easily adapt to seafood processing and storage methods such as modified atmosphere packaging, vacuum packaging, pH reduction, and the addition of sodium chloride (NaCl) [13,14].

Although OV50 can provide significant preservation benefits, as a lactic acid strain, OV50 is dependent on several parameters, such as temperature, humidity, and other environmental conditions, as well as the initial composition of the product being preserved, including its pH or initial microbial load. These variations could affect the strain’s ability to produce antimicrobial metabolites and, thus, the overall performance of the preservation process. The present study provides evidence that the newly isolated LAB OV50 can be used as a natural biopreservative to extend the shelf life of fresh sardines under refrigerated conditions by reducing the levels of TVB-N and PV concentrations, as well as the number of *Enterobacterales*, Coliforms, *Pseudomonas* spp., *Vibrio* spp., and *S. aureus*.

## 2. Material and Methods

### 2.1. Isolation and Selection of Lactic Acid Bacteria Strains

Seven samples of green olives in brine (Chemlal variety) were collected in December 2019 from five different regions of Bejaia (Algeria) as a showcase of traditional Algerian preparations. Then, 1 g of olive and 1 mL of brine from each sample were aseptically collected and added to 9 mL of de Man Rogosa and Sharpe (MRS) [15] broth (Conda, Madrid, Spain). After 18 h at 30 °C, we performed successive subcultures on MRS agar plates (Conda, Madrid, Spain) and incubated these plates at 30 °C for 48–72 h until pure colonies were obtained. Afterwards, the Gram-positive and catalase-negative isolates were presumed to be LAB strains. Their antibacterial activities were evaluated using the agar diffusion method [16]. To achieve this, we tested the cell-free supernatant (CFS) obtained by centrifuging (8000× *g*, 10 min, 4 °C) a fresh overnight LAB culture against *S. aureus* ATCC 6538 and *E. coli* ATCC 8739, which were used as target organisms. Next, CFS with the highest antibacterial activity was concentrated 10-fold using a Rotavapor (Stuart, Staffordshire, ST15 0SA, UK) and tested against the target organisms mentioned above; the isolate with the highest antibacterial activity was then selected for the rest of the work.

### 2.2. Strains Identification by API 50 System

OV50 was identified by using the API 50 CH kit from BioMérieux (Marcy l’Etoile, France) as recommended by the manufacturer. Briefly, 0.5 McFarland bacterial suspensions were inoculated into 50 microtubes of the API 50 CHL gallery and incubated at 37 °C for 24 to 48 h. The results were analyzed using the Apiweb™ Identification software (V5.1) on the website (https://apiweb.biomerieux.com accessed on 5 February 2022) [17].

### 2.3. Strains Identification by Mass Spectrometry and 16S rDNA Gene Sequencing

OV50 was identified using Maldi-TOF [18] and by 16S rDNA gene sequencing. For this purpose, we recovered total genomic DNA using the NucleoSpin Microbial DNA Mini Kit (Macherey-Nagel, Düren, Germany) and amplified the 16S rDNA gene using universal primers S1 (5′ AGAGTTTGATC(A,C)TGGCTCAG-3′) and S2 (5′GG(A,C)TACCTTGTTACGA(T,C)TTC-3′) and the PCR program previously reported [19]. The PCR products were electrophoresed on 2% (*w*/*v*) agarose gel for 2 h at a constant voltage of 75 V in Tris-Borate-EDTA buffer (TBE) (Sigma-Aldrich, Schnelldorf, Germany). Gels were stained using GelRed (Biotium, Fremont, CA, USA) and visualized using the GelDoc device (Bio-Rad, Hercules, CA, USA). The obtained sequences were analyzed using BLAST (Basic Local Alignment Search Tool) on the NCBI website (National Center for Biotechnology Information, http://www.ncbi.nlm.nih.gov accessed on 13 May 2022).

### 2.4. Safety Aspects

#### 2.4.1. Hemolytic Activity

The hemolytic activity of strain OV50 was characterized by streaking a fresh culture aliquot of 5 µL onto a Columbia agar horse blood plate (Liofilchem, Abruzzo, Italy) and incubating it at 30 °C for 48 h. Plates were examined for the presence of clear zones around colonies (β-hemolysis), green zones around colonies (α-hemolysis), or no hemolysis (γ-hemolysis) [20].

#### 2.4.2. Antibiotic Susceptibility

The antibiotic susceptibility of OV50 was determined using the VITEK 2 device system from bioMérieux (France). Therefore, seven antibiotics, including amoxicillin, clavulanic acid, ampicillin, imipenem, ciprofloxacin, vancomycin, erythromycin, and clindamycin, were utilized. Results were expressed in millimeters after measuring the diameters of the inhibition zones. The interpretation of the antibiogram was done according to EUCAST recommendations (European Committee on Antimicrobial Susceptibility Testing, http://www.eucast.org/ast_of_bacteria/disk_diffusion_methodology/, accessed on 8 January 2022).

#### 2.4.3. Cytotoxicity

The cytotoxicity of OV50 was evaluated across a Caco-2 intestinal epithelial cell line, grown to confluence in 96-well plates as previously reported [21]. OV50 was tested at 5 log and 6 log CFU/mL for a multiplicity of infection (MOI) of 1 and 10, respectively. Overnight cultures of OV50 incubated at 37 °C were centrifuged (8000× *g*, 10 min, 4 °C), washed twice with 4 mL of phosphate-buffered saline solution (PBS), and resuspended in Dulbecco’s modified Eagle’s medium (DMEM) (Thermo Fisher Scientific, Waltham, MA, USA), supplemented with 10% fetal bovine serum (Aidenbach, Germany) and without addition of antibiotics. Caco-2 cells were washed twice to completely remove the culture medium and then covered with 100 µL of bacterial suspension. DMEM was used as a negative control. Plates were incubated at 37 °C for 24 h and under 5% CO_2_. After incubation, the wells were washed twice to remove bacterial cells, and a solution of DMEM supplemented with 5% of CCK-8 reagent (Dojindo Molecular Technology, Tokyo, Japan) was added to each well. The plate was incubated again for 2 another hours, and the absorbance was measured in each well at 450 nm. Epithelial cell viability was calculated from the absorbance relative to the untreated control.

### 2.5. Technological Properties

The evaluation of the proteolytic and lipolytic activities of OV50 is critical prior to its application as a natural biopreservative for fresh sardines. These enzymatic activities not only contribute to the improvement of flavor and texture but also play a key role in lipid oxidation and protein degradation, ultimately improving the quality and shelf life of the preserved sardines.

#### 2.5.1. Proteolytic Activity

The proteolytic activity of OV50 was evaluated by examining bacterial cells obtained after 18 h of growth at 30 °C. These bacterial cells were washed three times with sterile saline solution and resuspended in the same buffer. Next, a 10 μL aliquot of the bacterial strain was inoculated onto nutrient agar (Conda, Madrid, Spain) supplemented with 10% (*m*/*v*) skimmed milk. After 48 h at 30 °C, proteolysis was observed by clear zones around the colonies [22].

#### 2.5.2. Lipolytic Activity

The lipolytic activity of OV50 was evaluated on MRS (pH 7) supplemented with 1% of olive oil and cod liver oil (fish oil). Plates were inoculated with 10 μL of OV50 suspension, washed three times with physiological sterile water, and resuspended in the same solution. After 48 h at 30 °C, the lipolysis was indicated by a clearing zone surrounded by the inoculated strains [23].

### 2.6. Application of Lactiplantibacillus plantarum OV50 for Biopreservation of Fresh Sardines

#### 2.6.1. Coculture between *Lactiplantibacillus plantarum* OV50 Alongside Pathogenic Strains in Sardine Culture Medium

*E. coli* ATCC 8739, *S. aureus* ATCC 6538, *V. cholerae* ATCC 14035, and *P. aeruginosa* ATCC 27853 were cultured with OV50 on a sardine-based medium, prepared as previously reported [24] with minor modifications. To prepare this medium, 100 g of sardines that have been washed, shelled, gutted, and boned were homogenized in 1 L of distilled water using an electric mincer. The mixture was then boiled for 5 min and filtered through Watman paper. The filtrate was supplemented with 2% glucose (Sigma-Aldrich, Schnelldorf, Germany) and autoclaved at 110 °C for 10 min; the addition of glucose is intended to provide a carbon and energy source to promote bacterial growth. Three cultures were prepared for the study. The first culture was a pure culture of the foodborne pathogen at 4 log CFU/mL. The second culture was a single culture of OV50 at 8 log CFU/mL. The third culture was a coculture of both microorganisms at the above-cited concentrations. These cultures were incubated at 37 °C and counted after 2, 4, 6, and 24 h on MRS for LAB, EMB agar (Liofilchem, Abruzzi, Italy) for *E. coli*, Baird Parker (Novachem, Quito, Ecuador) for *S. aureus* and Citrimid agar (Pasteur Institute, Algiers, Algeria) were used for *P. aeruginosa*, while peptone saline and alkaline agar (Biokar, Beauvais, France) were used for *V. cholerae*. The same experiments were then performed at 7 °C, and counts were performed every 48 h for 10 days.

#### 2.6.2. Application of *Lactiplantibacillus plantarum* OV50 as a Biopreservative Agent in Fresh Sardines

Two kilograms of fresh sardines were purchased early in the morning from the local market in the city of Bejaia (Algeria). The sardines were transported to the laboratory in a cooler and immediately cleaned, peeled, gutted, and boned into fillets with an average weight of 20 g each. The quantity was divided into two batches of 900 g each. A solution of 2 L of 4% glucose and 5% NaCl (solution A) was then prepared and autoclaved. One liter of sterile MRS was inoculated with 1% of an overnight culture of OV50 and incubated at 30 °C for 24 h. The culture was centrifuged at 8000× *g* for 10 min at 4 °C and washed twice with sterile saline water (0.9% NaCl). After that, it was reconstituted with the same volume of solution A. Sardines were inoculated and immersed in solution A containing the biopreservative OV50 strain at about 6 log CFU/g and were stored at 7 °C. Non-inoculated sardines were immersed in solution A-OV50 free and therefore served as the control. Afterwards, LAB, total mesophilic count (TMC), and naturally occurring spoilage microbial populations, such as *Vibrio* spp., *Pseudomonas* spp., *Enterobacterales*, Coliforms, and *S. aureus,* were determined at 0, 4, 8, and 12 days (coded D0, D4, D8, and D12, respectively) in both the control and inoculated sardines. Briefly, 10 g of each sample was taken aseptically and homogenized for 2 min in 90 mL of sterile saline water. Serial dilutions were prepared in the same diluent and plated onto appropriate plates. *Vibrio* were enumerated using peptone saline and alkaline agar. Coliforms, *Enterobacterales*, *Pseudomonas* spp., and *S. aureus* were also enumerated. *S. aureus*, LAB, and TMC were enumerated on VRBL agar (Liofilchem, Abruzzi, Italy), VRBG agar (Liofilchem, Abruzzi, Italy), Cetrimide agar, Baird Parker agar, MRS agar, and plate count agar (PCA) (Liofilchem, Abruzzi, Italy), respectively. The pH of the sardines was determined using a pH meter (HANNA Instruments, 584 East Drive, Woonsocket, RI, USA) after homogenizing 10 g of sardines in 90 mL of distilled water.

#### 2.6.3. Measurement of Peroxide Value

The peroxide value (PV) was determined using the official AOAC method [25]. To dissolve the fat, 7 g of sardine was ground and thoroughly stirred for 3 min with 30 mL of acetic acid-chloroform solution (Sigma-Aldrich, Schnelldorf, Germany) in a ratio (2:1 *v*/*v*). To the filtrate, we added 1 mL of saturated potassium iodide solution (Sigma-Aldrich, Schnelldorf, Germany) and stirred thoroughly for 3 min. After this, 70 mL of distilled water were added and shaken vigorously, and 1 mL of 1% starch solution (Sigma-Aldrich, Schnelldorf, Germany) was added. Titration was performed with sodium thiosulfate (Sigma-Aldrich, Schnelldorf, Germany) (N = 0.01) until the blue color disappeared. PV was calculated as milli-equivalents of peroxide per kilogram of sample (Formula (1)).
(1)$$\mathrm{PV}\;(\mathrm{meq}/\mathrm{kg})=\frac{V\;\times \;N}{W}\left(1000\right)$$Here, V is the volume of titration (mL), N is the normality of sodium thiosulfate solution (N = 0.01), and W the sample weight (kg).

#### 2.6.4. Determination of Total Volatile Basic Nitrogen (TVB-N)

The determination of TVB-N was performed according to the European Commission regulation [26]. Briefly, 5 g of sardines were ground and mixed with 45 mL of 6% perchloric acid solution (Sigma-Aldrich, Schnelldorf, Germany) in a suitable container. After homogenization for two minutes, the mixture was filtered. After the addition of 4 mL of 20% NaOH solution containing three drops of phenolphthalein (1%), 25 mL of the extract was steam distilled. The distillate was collected in a receiver containing 50 mL of 3% boric acid solution (Sigma-Aldrich, Schnelldorf, Germany) to which three to five drops of Tashiro mixed indicator solution (methyl red/methylene blue, 0.2/0.1%) were added. After distillation, the volatile bases in the receiver solution were measured by titration with standard hydrochloric acid solution (0.01 M) until the pH reached 5.0 ± 0.1. TVB-N was calculated according to Formula (2).
(2)$$\mathrm{TVB-N}\;(\mathrm{mg}/100\;g\;\mathrm{sample})\;=\frac{(V1\;-\;V0)\;\times \;0.14\;\times \;2\;\times \;100}{M}$$
where TVB-N is a total volatile basic nitrogen, V1 is a volume of 0.01 M hydrochloric acid solution in mL for sample, V0 is a volume of 0.01 M HCl solution in mL for blank, and M is a weight of sample in grams. Blank: 45 mL perchloric acid solution (6%) was used instead of the extract.

### 2.7. Statistical Analyzes

Data from all analyzes were collected, statistically analyzed, and expressed as mean ± SD (n = 3). Statistica^®^ software version 8 (Inc. StatSoft, Tulsa, OK, USA) was used for this study. ANOVA was performed using the least significant difference (LSD) test, and Student’s *t*-test was used to analyze differences between two variables. Results were considered statistically significant at a *p*-value < 0.05, which was chosen as the minimum level of significance.

## 3. Results

### 3.1. Lactiplantibacillus plantarum OV50 Revealed Highest Antibacterial Activity

Seventeen Gram-positive bacilli deprived of catalase activity were isolated from seven different samples of olives in brine, as shown in Figure 1. All CFS were active against *E. coli* ATCC 8739 and *S. aureus* ATCC 6538, especially those prepared from strains 44, 50, 51, and 58. The antibacterial activity increased with the concentration of the CFS, as shown in Figure 2. It is noteworthy that none of the supernatants showed antibacterial activity after pH neutralization of the CFS. After testing the antibacterial activity against *S. aureus* ATCC 6538 and *E. coli* ATCC 8739, OV50 was selected as the best candidate for further characterization and application.

### 3.2. Molecular Identification

The API50 galleries are a standardized system consisting of 50 tests; this enables us to establish the carbohydrate fermentation profile. The API50 system (https://apiweb.biomerieux.com accessed on 1 February 2022) permitted to identify OV50 at the genus level and its complete identification as *Lactiplantibacillus plantarum* was achieved using MALDI-TOF and 16S rDNA gene sequencing methods.

### 3.3. Lactiplantibacillus plantarum OV50 Is Safe

The safety of OV50 was evaluated by analyzing its hemolytic activity, antibiotic resistance, and cytotoxicity against eukaryotic Caco-2 intestinal epithelial cells. When fresh colonies of OV50 were streaked onto a blood agar plate, no hemolysis was observed under the conditions studied. In addition, OV50 has been tested for susceptibility to different antibiotics and the results are expressed in millimeters (Table 1). OV50 was found to be sensitive to most of the antibiotics tested, except for vancomycin and ciprofloxacin, which are intrinsically resistant. Our results indicate that OV50 does not exhibit cytotoxicity against Caco-2 cells. The absorbance at 450 nm using the CCK8 assay was similar to that of the control group, and the percentage of cell viability was 100%, indicating that the cells remained viable and were not adversely affected by the presence of OV50.

### 3.4. Lactiplantibacillus plantarum OV50 Is Not Proteolytic Nor Lipolytic

Proteolytic and lipolytic activities were evaluated and both were found to be negative. This is probably due to the absence of lipolytic and proteolytic enzymes.

### 3.5. Coculture Results

Cocultures were performed using a sardine-based culture medium. As a result, at 37 °C, the growth of *V. cholerae* ATCC 14035 and *P. aeruginosa* ATCC 27835 were inhibited after 4 h of coculturing with OV50. Similarly, the growth of *S. aureus* ATCC 6538 and *E. coli* ATCC 8739 were inhibited after 6 h of coculturing with OV50. These results were significantly different (*p* < 0.05) from those obtained with pure cultures. The number of foodborne pathogens in individual cultures varied from 4 to 8 log CFU/mL for each pathogen (Figure 3).

OV50 exhibited bactericidal activity against these pathogens at 37 °C. The growth of OV50 remained stable (8 log CFU/mL) during the incubation period in pure cultures and cocultures at 37 °C, as shown in Figure 3. Additionally, the pH level of the pathogen’s cultures remained stable at pH 6 for the first 6 h of incubation. After 24 h, however, the pH level noticeably decreased to 4.5 for both *E. coli* ATCC 8739 and *S. aureus* ATCC 6538. For *V. cholerae* ATCC 14035, the pH level decreased to pH 5 but remained stable thereafter, with no significant difference (*p* > 0.05) observed for *P. aeruginosa* ATCC 27835 (Figure 4).

In pure culture, OV50 was able to lower the pH to 3. In coculture, the pH followed a similar decrease pattern with no significant difference compared to the pure culture of OV50. The coculture was also performed in the refrigerator to study the interaction of OV50 with pathogens at low temperatures. Figure 5 shows that OV50 completely inhibited the growth of *E. coli* ATCC 8739 and *V. cholerae* ATCC 14035 after 8 days of incubation. It also inhibited the growth of *P. aeruginosa* ATCC 27835 after 6 days but only reduced the growth of *S. aureus* ATCC 6538 from 4 log to 2 log CFU/mL after 10 days. These results demonstrate a bactericidal effect on *E. coli* ATCC 8739, *V. cholerae* ATCC 14035, and *P. aeruginosa* ATCC 27835, and a bacteriostatic effect on *S. aureus* ATCC 6538. The growth of *E. coli* ATCC 8739, *V. cholerae* ATCC 14035, and *S. aureus* ATCC 6538 remained stable at 4 log CFU/mL. However, *P. aeruginosa* ATCC 27835 showed a significant increase (*p* < 0.05) from 4 to 5 log CFU/mL (Figure 5). The growth of OV50 alone or in coculture showed no significant difference (*p* > 0.05) and remained stable at 8 log CFU/mL (Figure 5). The pH of the pathogen culture remained stable at pH 6, while the pH of OV50 decreased to 3.5 when grown alone or in coculture with pathogens (Figure 6). These experiments demonstrate that the decrease in pH in pure and coculture conditions was primarily influenced by OV50.

### 3.6. Microbial Counts in Preserved Sardines

Figure 7 shows a significant difference (*p* < 0.05) between sardines inoculated with OV50 and the control group during storage. LAB counts increased from 4 CFU/g to 4.55 log CFU/g in the control samples and from 6.42 log CFU/g to 7.31 log CFU/g in the samples inoculated with OV50 (Figure 7a). In addition, the control group had a TMC of 2.76 log CFU/g at the start of storage (Figure 7b). The count showed a significant (*p* < 0.05) increasing trend over time, reaching 5.93 log CFU/g at the end of storage. After D12 of storage, the TMC of sardines inoculated with OV50 was 6.41 log CFU/g compared to 5.48 log CFU/g at the beginning of storage. Throughout the storage period, there was a significant difference (*p* < 0.05) between the control group and the inoculated group, with the highest value observed in the inoculated group. During storage at 7 °C, the number of *Enterobacterales* increased significantly (*p* < 0.05). In the control group, the counts increased from 2.18 to 4.25 log CFU/g, and in the inoculated group, they increased from 2.06 to 3.69 log CFU/g (Figure 7c). At D8 and D12, *Enterobacterales* counts were significantly lower in the inoculated group than in the control group (*p* < 0.05). In addition, the levels of coliforms (Figure 7d) increased significantly from 1.87 to 4.43 log CFU/g and from 1.85 to 3.78 log CFU/g for the control and inoculated groups, respectively. A statistically significant difference (*p* < 0.05) was observed between the two groups at D8 and D12. Figure 7e shows a significant increase (*p* < 0.05) in *Pseudomonas* counts in both groups, ranging from 1.5 CFU/g to 3.17 log CFU/g in the control group and 1 CFU/g to 1.74 log CFU/g in the inoculated group. In addition, a significant difference (*p* < 0.05) was observed between the two groups at D8 and D12, with a reduced rate in the inoculated group compared to the control group. In the control group, *Vibrio* levels increased significantly (*p* < 0.05) from 2.50 to 4.82 log CFU/g at D12. In the inoculated group, levels increased from 2.56 to 4.08 log CFU/g at D4, then decreased to 3.22 log CFU/g at D8, with a significant difference (*p* < 0.05) compared to the control group. Of note, *Vibrio* levels slightly increased at D12 (3.66 log CFU/g) but with no significant difference (*p* > 0.05) from D8 (Figure 7f). Comparable results were also obtained for *S. aureus* (Figure 7g). The bacterial counts significantly increased (*p* < 0.05) from 1.52 to 4.79 log CFU/g in the control group and from 1.69 to 2.86 log CFU/g in the inoculated group during the 12 days of storage. However, there was a significant decrease (*p* < 0.05) in D8, where the counts decreased from 2.9 to 2.26 log CFU/g in the inoculated group.

### 3.7. Physicochemical Results

#### 3.7.1. Peroxide Value

Figure 8a shows that in the sardine control group, PV increased from 5.72 ± 1.43 on the first day to 11.44 ± 1.43 after 4 days of storage (*p* < 0.05), before decreasing to zero after D8 and D12. The amount of peroxide in sardines inoculated on the first and D4 did not change significantly (*p* > 0.05) (5.9 ± 0.44 and 6.6 ± 1.09, respectively). According to Ghomi et al. [27], fish is safe for consumption if it contains 10–20 peroxide per kg of fish fat. Significant differences (*p* < 0.05) were observed between the control and experimental groups after four days of storage.

#### 3.7.2. pH Monitoring

The initial pH of the OV50-treated group was 6.16 ± 0.01, while the untreated group had a pH of 6.07 ± 0. After D8 of storage, the pH of the OV50 treated group decreased to 4.79 ± 0.07 and then increased to 5.33 ± 0.2 at the end of storage (D12). In contrast, the control group had a pH of 7.96 ± 0.19 at the end of storage (Figure 8b). Significant differences (*p* < 0.05) were observed between the treated and control groups throughout the 12-day storage period. The use of OV50 in the preservation of sardines significantly lowered the pH, allowing for stable storage conditions.

#### 3.7.3. TVB-N

The results shown in Figure 8c indicate that the initial TVB-N concentration was 2.9 ± 0.85 mg/100 g on the first day of storage and remained below 35 mg/100 g for the inoculated group for the next 8 days. The acceptable limit of 35 mg/100 g, according to the decision of the European Commission [28], was not exceeded before 8 days of storage, but it was exceeded on day 12, resulting in a value of 46.66 mg/100 g of sardines. Therefore, the chilled fish were edible for up to 8 days of storage. In the control group, the permissible level of TVB-N was exceeded on the 8th day of storage, resulting in a value of 59.77 ± 8.4 and 79.52 ± 8.4 mg/100 g after 12 days of storage. Throughout the storage period (12 days), significant differences (*p* < 0.05) were observed between the inoculated and control groups of sardines.

## 4. Discussion

The selection of LAB strains as natural biopreservatives is based on their inhibitory properties [29]. We isolated 17 novel antagonistic LAB strains from traditional Algerian olive preparations and related their inhibitory properties to the production of organic acids, especially lactic acid since the isolates were homofermentative. Due to its remarkable *in vitro* antibacterial activity, we selected strain OV50 for further characterization and application as a natural biopreservative for fresh sardines. OV50 is a non-hemolytic and non-cytotoxic strain that is intrinsically resistant to antibiotics, which is typical of LAB and does not preclude further exploration of this strain [30,31]. These results support the use of OV50 as a biopreservative in fresh sardines. Proteins have a major impact on the texture and nutritional value of sardines [32], while lipids are capable of causing rancidity and degrading the flavor of foods [33]. Maintaining these natural protein and lipid levels throughout product storage is expected to be an added value, which was ensured by the OV50 strain used in this study. However, lipid oxidation may occur in sardine muscle during storage due to high concentrations of unsaturated fatty acids [34,35]. The null PV obtained after D8 of storage could be attributed to hydroperoxide degradation, which decomposes into various metabolic products, such as aldehydes [36]. It is worth noting that inhibition of lipid peroxidation in sardines by LAB has been reported [37,38,39]. Seafood is highly perishable due to the presence of undesirable breakdown compounds such as trimethylamine, dimethylamine, methylamine, and ammonia, collectively known as TVB-N. The levels of these compounds indicate the deterioration of the seafood [40,41,42]. The increase in TVB-N in sardine fillets is directly associated with proteolysis and degradation of nitrogenous and volatile compounds by spoilage microorganisms [42,43]. The use of OV50 significantly reduced the TVB-N level in fresh sardines, which is consistent with previously reported data [44,45]. OV50’s ability to reduce TVB-N levels in fresh sardines demonstrates its effectiveness in preserving seafood quality by inhibiting protein degradation and minimizing spoilage indicators. This is in line with the broader goal of extending shelf life, preserving nutritional value, and meeting consumer expectations for fresh and high-quality seafood products.

The antibacterial properties of OV50 were evaluated *in situ* on a sardine medium, at temperatures of 7 and 37 °C, when it was intentionally inoculated in coculture with food-borne pathogens. The growth stability of OV50 (8 log CFU/mL), at both temperatures (7 and 37 °C) and throughout the incubation period, argues that sardines are conducive to the survival of this strain and supports the ability of this selected strain to grow at low temperatures. The stability observed in the presence of foodborne pathogens could be due to the amensalism effect, i.e., the growth of the lactic strain was not affected and reduced by the pathogenic strains [46,47].

Inhibition of foodborne pathogens by organic acids is a critical factor in inhibiting their growth, as they thrive at a neutral pH [48,49]. LABs produce organic acids that act on the cell membranes of harmful strains and interfere with nutrient metabolism [50].

In particular, the foodborne pathogens *E. coli* ATCC 8739, *V. cholerae* ATCC 14035, and *S. aureus* ATCC 6538 showed a moderate decrease in pH at 37 °C, which could be explained by their ability to metabolize medium components and produce organic acids [51,52,53]. In the case of *P. aeruginosa* ATCC 27835, the pH stability of the pure culture could be explained by the metabolic plasticity of this strain, which could trigger metabolic pathways that do not necessarily lead to significant acidification [54]. However, the pH stability of pure cultures of these foodborne pathogens at 7 °C compared to 37 °C suggests a possible influence of temperature on the metabolic activity of these microorganisms. Their metabolism is likely reduced by the lower temperatures, which may affect their metabolite production and, consequently, the pH of the medium. Other studies also supported the inhibitory properties of LAB in coculture against foodborne pathogens [46,47,55,56,57]. Environmental factors have been reported as the main sources of contamination of fish products [4]. LAB can shield seafood from these microbial contaminations and alterations by producing metabolites endowed with inhibitory activities or by nutritional competition [37,58]. Here, the high number of LAB observed in the inoculated group could be attributed to the growth of OV50, while the LAB growth reported in the control group could be explained by endogenous LAB, as yet reported [38,59,60]. The bioprotective potential of endogenous LAB against pathogens has been highlighted. An increasing number of studies are aimed at exploiting this ability to control the quality and safety of marine products, and some industries producing LAB starters are currently testing some bacteria for a fish application [61]. During the storage period, we noted that the TMC in the control group was significantly close to the suggested limit of log 5.70 CFU/g [62], while that found in the inoculated group exceeded this value explainable by the presence of OV50 reported to grow in PCA. It is noteworthy that the counts of *Enterobacterales*, Coliforms, *Pseudomonas* spp., *Vibrio* spp., and *S. aureus* at 7 °C were reduced by the presence of OV50, which produces organic acids and inhibits the growth of bacteria, other than LAB [39]. The results obtained here are consistent with those previously reported [56,57,58,63]. pH reduction is a critical parameter in food storage management, as it acts as a barrier to spoilage microorganisms [64]. Interestingly, OV50 induced a significant decrease in pH, allowing for proper storage of fresh sardines. The increase in pH in the control group could be due to the growth of alkalinizing bacteria and the formation of basic nitrogenous substances [65,66,67]. There are apparently no pH recommendations for fish products [66].

## 5. Conclusions

The recently isolated OV50 strain is safe and can protect fresh sardines from *Enterobacterales*, Coliforms, *Pseudomonas* spp., *Vibrio* spp., and *S. aureus* during cold storage. The OV50 strain not only has relevant inhibitory properties but also affects the formation of total VNB and peroxide in sardines. Overall, these results confirm that OV50 is a useful natural biopreservative that can reduce the risk of spoilage and pathogenic microorganisms in sardines, thereby reducing the production of negative compounds.

## Figures and Tables

**Figure 1 foods-13-00368-f001:**
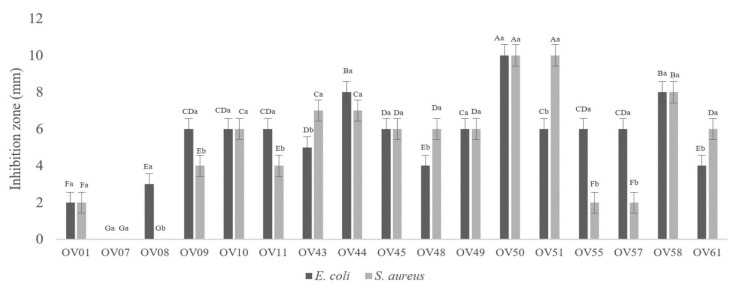
Antibacterial activity of native supernatant of isolates against *E. coli* ATCC 8739 and *S. aureus* ATCC 6538. A–G: significant difference (*p <* 0.05) between activities of isolates, a,b: significant difference for each isolate against the two pathogens.

**Figure 2 foods-13-00368-f002:**
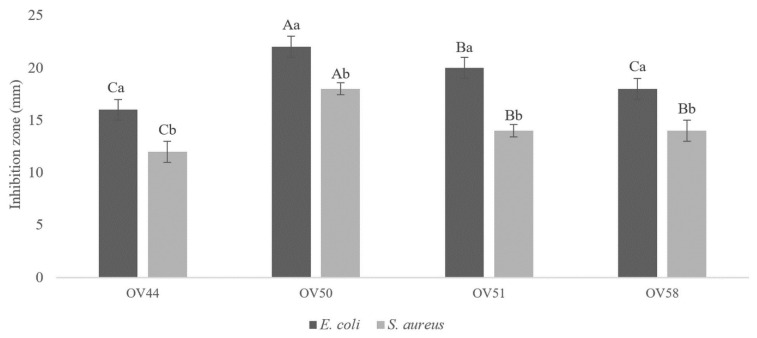
Antibacterial activity of concentrated supernatant of isolates against *E. coli* ATCC 8739 and *S. aureus* ATCC 6538. A–C: significant difference (*p <* 0.05) between activities of isolates, a,b: significant difference for each isolate against the two pathogens.

**Figure 3 foods-13-00368-f003:**
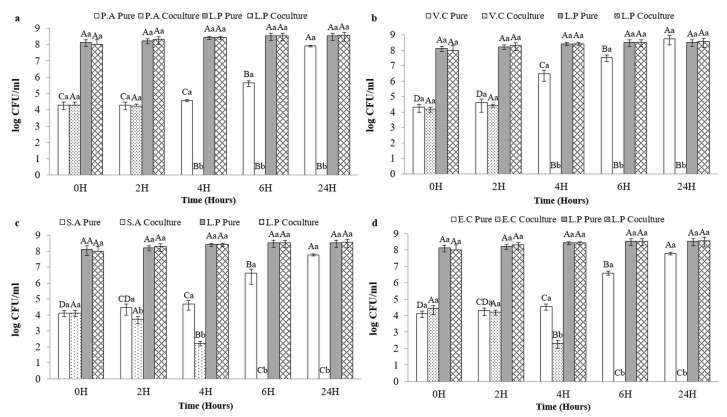
Effect of *L. plantarum* OV50 (L.P) in coculture with: (**a**) *P. aeruginosa* ATCC 27835 (P.A), (**b**) *V. cholerae ATCC* 14035 (V.A), (**c**) *S. aureus* ATCC 6538 (S.A), and (**d**) *E. coli* ATCC 8739 (E.C) at 37 °C. A–D: significant difference (*p <* 0.05) during time, a,b: significant difference between the inoculated and the control group of sardines.

**Figure 4 foods-13-00368-f004:**
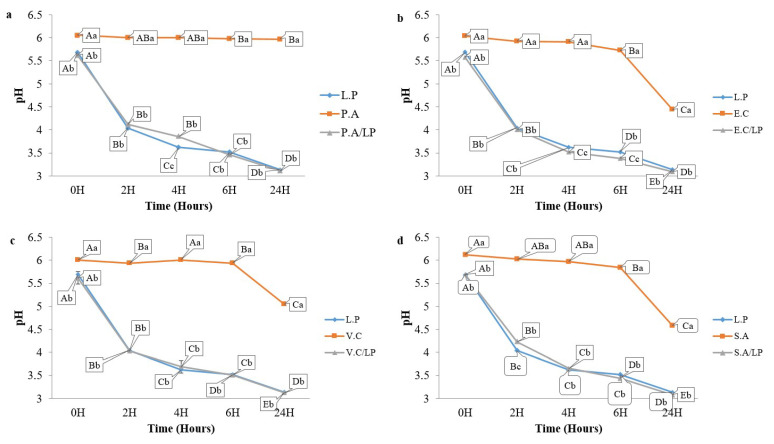
pH evolution during coculture: (**a**) *P. aeruginosa* ATCC 27835/*L. plantarum* OV50 (P.A/L.P), (**b**) *E. coli* ATCC 8739/*L. plantarum* OV50 (E.C/L.P), (**c**) *V. cholerae ATCC* 14035/*L. plantarum* OV50 (V.C/L.P) and (**d**) *S. aureus* ATCC 6538/*L. plantarum* OV50 (S.A/L.P) at 37 °C. A–E: significant difference (*p <* 0.05) during time, a–c: significant difference between the inoculated and the control group of sardines.

**Figure 5 foods-13-00368-f005:**
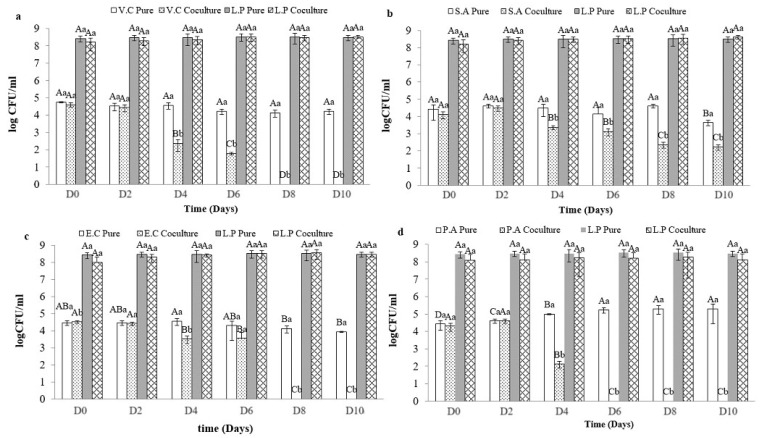
Effect of *L. plantarum* OV50 (L.P) in coculture with: (**a**) *V. cholerae* ATCC 14035 (V.A), (**b**) *S. aureus* ATCC 6538 (S.A), (**c**) *E. coli* ATCC 8739 (E.C), and (**d**) *P. aeruginosa* ATCC 27835 (P.A) at 7 °C. (A–D) significant difference (*p <* 0.05) during time, a,b: significant difference between the inoculated and the control group of sardines.

**Figure 6 foods-13-00368-f006:**
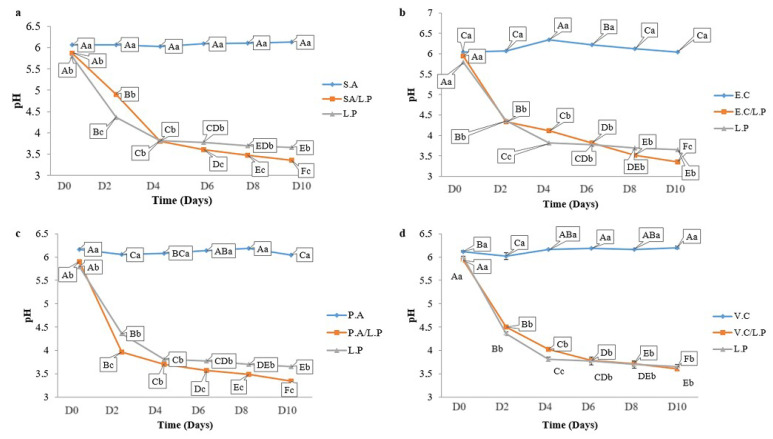
pH evolution during coculture: (**a**) *S. aureus* ATCC 6538/*L. plantarum* OV50 (S.A/L.P), (**b**) *E. coli* ATCC 8739/*L. plantarum* OV50 (E.C/L.P), (**c**) *P. aeruginosa* ATCC 27835/*L. plantarum* OV50 (P.A/L.P), and (**d**) *V. cholerae ATCC* 14035/*L. plantarum* OV50 (V.C/L.P) at 7 °C. A–F: significant difference (*p* < 0.05) over time, a–c: significant difference between inoculated and control group of sardines.

**Figure 7 foods-13-00368-f007:**
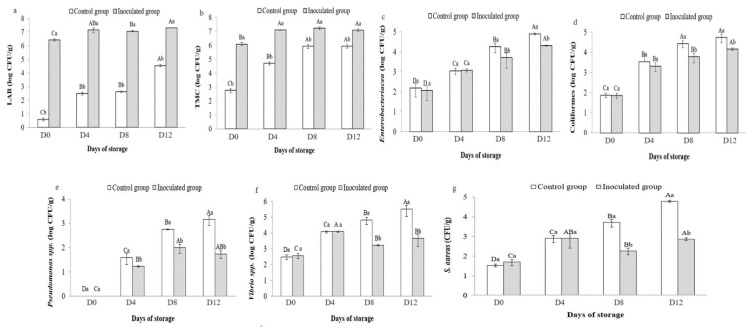
Enumeration of (**a**) lactic acid bacteria (LAB), (**b**) total mesophilic count (TMC), (**c**) *Enterobacterales*, (**d**) Coliforms, (**e**) *Pseudomonas* spp., (**f**) *Vibrio* spp., and (**g**) *S. aureus* in control and inoculated sardines. A–D: significant difference (*p <* 0.05) over time, a,b: significant difference between the inoculated and the control group of sardines.

**Figure 8 foods-13-00368-f008:**
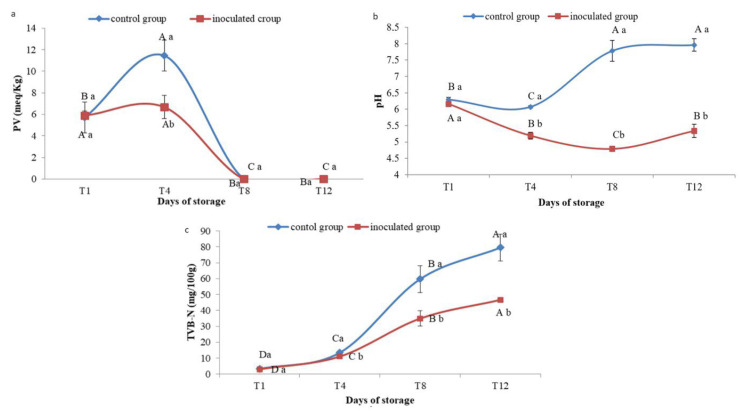
Changes in (**a**) peroxide value, (**b**) pH, and (**c**) total volatile basic nitrogen (TVB-N) of inoculated and control groups of sardines during storage. A–D: significant difference (*p* < 0.05) over time, a,b: significant difference between inoculated and control group of sardines.

**Table 1 foods-13-00368-t001:** Antibiotic spectrum of *Lactiplantibacillus plantarum* OV50.

Strain	Antibiotic Susceptibility (mm)						
	Ampi.	Amx. + Clavu. ac.	Imipen.	Cipro.	Vanco.	Erythro.	Clinda.
OV50	22	29	30	0	0	35	15

Amx.: amoxicillin, Clavu. ac.: clavulanic acid, Ampi.: ampicillin, Imipen.: imipenem, Cipro.: ciprofloxacin, Vanco.: vancomycin, Erythro.: erythromycin, and Clinda.: clindamycin.

## Data Availability

The original contributions presented in the study are included in the article, further inquiries can be directed to the corresponding authors.

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
