# Peer review of "Biopreservation of Fresh Sardines (Sardina pilchardus) Using Lactiplantibacillus plantarum OV50 Isolated from Traditional Algerian Green Olives Preparations"

_foods, 2024, doi:10.3390/foods13030368_

Round 1
Reviewer 1 Report
Comments and Suggestions for Authors
This work holds paramount significance as it introduces Lactiplantibacillus plantarum OV50 as a novel and safe biopreservative, showcasing its remarkable ability to inhibit spoilage and pathogenic microorganisms in fresh sardines. The findings offer promising implications for enhancing the shelf life and safety of seafood products, addressing critical challenges in the preservation of perishable foods and ensuring their quality for consumers.
Comments
Abstract
Provide specific numerical data for microbiological counts, TVB-N levels, and PV concentrations to offer a clearer understanding of the observed changes. For example, mention the actual values of microbial counts and the extent of reduction in TVB-N and PV.
Introduction
(Line 59): Emphasize how the use of LAB as a natural biopreservative aligns with consumer preferences for freshness, naturalness, and environmental concerns.
(Line 63): Clarify how LAB metabolites, like bacteriocins and organic acids, contribute to inhibitory properties against spoilage and pathogens.
(Line 74): Include specific data on the shelf life extension achieved by LAB OV50 in preserving fresh sardines under refrigerated conditions.
(Line 75): Briefly mention potential challenges or limitations in applying LAB OV50, such as variations in efficacy under different storage conditions.
Methods
(Line 118): Specify the units for antibiotic susceptibility results obtained using the VITEK 2 system. Include whether the values are expressed in millimeters or another unit for clarity.
(Line 125): Provide specific quantitative results regarding the cytotoxicity of OV50 on the Caco-2 intestinal epithelial cell line, such as the percentage of cell viability at different concentrations.
(Line 141): why the proteolytic and lipolytic activities of OV50 are relevant in the context of its application as a natural biopreservative for fresh sardines.
(Line 159): Elaborate on the composition and preparation of the sardine-based medium, including the rationale for incorporating glucose and the specific modifications made to the reported method
Results
(Line 254): Clarify the specific criteria or parameters used by the API50 system for identifying OV50 at the genus level.
(Line 260): Provide details on the specific antibiotics included in the susceptibility testing, indicating whether the reported values in Table 1 are expressed in millimeters or another unit.
(Line 282): Explain the reasoning behind selecting 4 h and 6 h as the time points for assessing the inhibitory effects on different pathogens during coculture.
(Line 289): Specify the units for microbial counts, particularly the log CFU/mL measurements, in Figures 3 and 5.
(Line 383): Specify the unit of measurement for total volatile basic nitrogen (TVB-N) in Figure 8c.
Discussion and Conclusion
(Line 398): Clarify the specific organic acids produced by OV50 that contribute to its inhibitory properties against spoilage and pathogenic microorganisms.
(Line 404): Expand on the concept of amensalism (Lines 424-426) and its role in the stability observed in OV50's co-culture with foodborne pathogens.
(Line 410): Discuss the potential impact of the null peroxide value (PV) obtained after D8 of storage (Lines 411-413) on the overall quality and safety of preserved sardines. Address the significance of PV breakdown products, such as aldehydes, and their role in maintaining sardine quality.
(Line 420): Elaborate on how OV50's reduction of total volatile basic nitrogen (TVB-N) levels in fresh sardines (Lines 419-420) aligns with the broader goal of preserving seafood quality.
(Line 422): Discuss the significance of OV50's ability to grow and inhibit foodborne pathogens at both 7°C and 37°C (Lines 422-423).
(Line 434): Discuss the observed pH stability of foodborne pathogens in pure culture at 7°C compared to 37°C (Lines 434-435).
(Line 442): Address the potential sources of endogenous LAB growth in the control group during the storage period (Lines 442-443). Provide insights into the factors contributing to LAB growth in the absence of inoculated OV50 and their implications for sardine preservation.
Comments on the Quality of English LanguageModerate editing of English language required
Author Response
After revising the manuscript and following meticulously, your appreciated remarks are included in the revised manuscript and highlighted in yellow. Thank you
Abstract
- Provide specific numerical data for microbiological counts, TVB-N levels, and PV
concentrations to offer a clearer understanding of the observed changes. For example,
mention the actual values of microbial counts and the extent of reduction in TVB-N and PV.
This is done for TVB-N and PV. For microbial count, there is a lot of data to present, if we mention it, the abstract will be too long.
Introduction
- (Line 59): Emphasize how the use of LAB as a natural biopreservative aligns with consumer preferences for freshness, naturalness, and environmental concerns.
Done (Please see line 65).
- (Line 63): Clarify how LAB metabolites, like bacteriocins and organic acids, contribute to inhibitory properties against spoilage and pathogens.
Done (Please see line 72)
- (Line 74): Include specific data on the shelf life extension achieved by LAB OV50 in preserving fresh sardines under refrigerated conditions
Done (Please see line 90)
- (Line 75): Briefly mention potential challenges or limitations in applying LAB OV50, such as variations in efficacy under different storage conditions.
Done (Please see line 83).
Methods
- (Line 118): Specify the units for antibiotic susceptibility results obtained using the VITEK 2 system. Include whether the values are expressed in millimeters or another unit for clarity.
Done (Please see line 138).
- (Line 125): Provide specific quantitative results regarding the cytotoxicity of OV50 on the Caco-2 intestinal epithelial cell line, such as the percentage of cell viability at different concentrations.
Done in result section (Please see line 293).
- (Line 141): why the proteolytic and lipolytic activities of OV50 are relevant in the context of its application as a natural biopreservative for fresh sardines.
Done (Please see line 160).
- (Line 159): Elaborate on the composition and preparation of the sardine-based medium, including the rationale for incorporating glucose and the specific modifications made to the reported method
Done (Please see line 187), the modification is the adding of glucose.
Results
- (Line 254): Clarify the specific criteria or parameters used by the API50 system for identifying OV50 at the genus level.
Done (Please see line 279)
- Provide details on the specific antibiotics included in the susceptibility testing, indicating whether the reported values in Table 1 are expressed in millimeters or another unit.
Done for the units (Please see line 289).
- (Line 282): Explain the reasoning behind selecting 4 h and 6 h as the time points for assessing the inhibitory effects on different pathogens during coculture.
The inhibitory effect is followed for 24 hours in sardine-based medium, point 4 and 6h correspond to the point where the 0V50 completely inhibited the food-borne pathogens.
- (Line 289): Specify the units for microbial counts, particularly the log CFU/mL measurements, in Figures 3 and 5.
- (Line 383): Specify the unit of measurement for total volatile basic nitrogen (TVB-N) in Figure 8c.
The figures have been verified, and the units are indicated on each figure.
Discussion and Conclusion
- Clarify the specific organic acids produced by OV50 that contribute to its inhibitory properties against spoilage and pathogenic microorganisms.
Done (Please see, line 439).
- (Line 404): Expand on the concept of amensalism (Lines 424-426) and its role in the stability observed in OV50's co-culture with foodborne pathogens.
Done (Please see, line 471).
- (Line 410): Discuss the potential impact of the null peroxide value (PV) obtained after D8 of storage (Lines 411-413) on the overall quality and safety of preserved sardines. Address the significance of PV breakdown products, such as aldehydes, and their role in maintaining sardine quality.
The zero peroxide value obtained after D8 does not mean that the oxidation processes have stopped, it is simply the result of the hydroperoxides breaking down into other harmful products, such as aldehydes, which are not detectable by the method used to evaluate the peroxide value. The formation of aldehydes is undesirable but unavoidable over the course of storage due to the oxidation of lipids.
- (Line 420): Elaborate on how OV50's reduction of total volatile basic nitrogen (TVB-N) levels in fresh sardines (Lines 419-420) aligns with the broader goal of preserving seafood quality.
Done (Please see, line 460).
- (Line 422): Discuss the significance of OV50's ability to grow and inhibit foodborne pathogens at both 7°C and 37°C (Lines 422-423).
Done (Please see, line 465).
- (Line 434): Discuss the observed pH stability of foodborne pathogens in pure culture at 7°C compared to 37°C (Lines 434-435).
Done (Please see, line 481).
- (Line 442): Address the potential sources of endogenous LAB growth in the control group during the storage period (Lines 442-443). Provide insights into the factors contributing to LAB growth in the absence of inoculated OV50 and their implications for sardine preservation.
Done (Please see, line 493).
Reviewer 2 Report
Comments and Suggestions for Authors
This manuscript proposes the use of the OV50 strain to preserve fresh Sardines. Given the economic importance and nutritional value of this seafood product, the purpose is very interesting, also because it is an alternative to other less sustainable and expensive methods (cold, radiation). However, the main gap between purpose and results is that there is no comparison (microbiological, sensorial, organoleptic) between sardines preserved with traditional methods and the method proposed by the authors. In detail:
Introduction: I would recommend writing using shorter sentences. for example from line 57 to line 62 the period is too long.
Results: the sequence of activities and analyzes is unclear. I imagine that in order to be able to attribute the strains to the species to which they belong, the molecular analyzes were carried out before the selection of the isolates, otherwise it is not clear why the best performing strain already has the species attribution from the first paragraph. I recommend tidying up.
The selection of the OV50 strain was made only on the basis of its antimicrobial activity and I ask myself: why not also characterize the other isolates, for example OV58, and others for their safety and technological characteristics? In fact, the antimicrobial activity in coculture is considered but only for pathogenic strains. However, the antimicrobial activity of multiple isolated strains in coculture could also be of interest.
Where and how are these strains stored?
Why was the ability of the isolates to produce biogenic amines not evaluated as a further selection parameter?
All figures require greater clarity of information and all legends are not completely informative.
How can you be sure that the antimicrobial activity was performed by the OV50 strain in the inoculated sardines? It could be useful to develop a molecular method for monitoring the OV50 strain such as specific primers PCR.
The comparison between the various methods is missing. However, is also very important from a sensorial point of view, at least with a liking test following the comparison with sardines subjected to other preservation methods.
Author Response
After revising the manuscript and meticulously following your appreciated remarks are included below and highlighted in bleue color in the text.
Introduction: I would recommend writing using shorter sentences. for example from line 57 to line 62 the period is too long.
Done, lines 57 through 62 reformulated with a short sentence (Please see, line 59).
Results: the sequence of activities and analyzes is unclear. I imagine that in order to be able to attribute the strains to the species to which they belong, the molecular analyzes were carried out before the selection of the isolates, otherwise it is not clear why the best performing strain already has the species attribution from the first paragraph. I recommend tidying up.
The title "Isolation and characterization of Lactiplantibacillus plantarum OV50" is replaced by the title "Isolation and selection of lactic strains" (line 94) to clarify the steps of the work, which are as follows:
We started with the isolation of 17 isolates from olive brine. Then we carried out a screening based on antibacterial activity. Since the context of the work focuses on the use of lactic acid bacteria to reduce microbial spoilage of fresh sardines, we considered antibacterial activity as the most important selection criterion. Finely we identified the isolate jug with the best antibacterial activity as "Lactiplantibacillus plantarum OV50" and we use it for the rest of the work.
The selection of the OV50 strain was made only on the basis of its antimicrobial activity and I ask myself: why not also characterize the other isolates, for example OV58, and others for their safety and technological characteristics?
Yes, your suggestion is very interesting, but we felt that it was prudent to characterize other strains and use them in separate studies, given the wealth of results in this particular study.
In fact, the antimicrobial activity in coculture is considered but only for pathogenic strains. However, the antimicrobial activity of multiple isolated strains in coculture could also be of interest.
Yes indeed, the study of antimicrobial activity of multiple isolated strains in coculture could also be of interesting but this is not exactly the aim of this study.
Where and how are these strains stored?
The lactic acid strains were stored in MRS bags and the pathogenic strains were stored in nutrient bags at -80°C.
Why was the ability of the isolates to produce biogenic amines not evaluated as a further selection parameter?
We agree with the referee that this point is relevant. It is expected that the whole genome of this strain will be fully sequenced and analysed for bacteriocin DNA determinants as well as for decarboxylases. If such genetic determinants are found, further studies on amine biogenes will be performed.
All figures require greater clarity of information and all legends are not completely informative.
Some details are added to the legends (OV50 is previoused by L. plantarum to clarify the abbreviation L.P) Please see, lines (313, 331,332,333,354,358,359,360).
How can you be sure that the antimicrobial activity was performed by the OV50 strain in the inoculated sardines? It could be useful to develop a molecular method for monitoring the OV50 strain such as specific primers PCR.
We attribute that the antimicrobial activity was carried out by the OV50 strain in the inoculated sardines, by comparing with the results of the control group, seeing that the inoculated group is more protected than the control one, explain that the antibacterial activity is due to the presence of the OV50 strain and not to the endogenous bacteria of sardine. The molecular method PCR is expected to be established after the complete sequencing and exploitation of the genome of the strain.
The comparison between the various methods is missing. However, is also very important from a sensorial point of view, at least with a liking test following the comparison with sardines subjected to other preservation methods.
Thanks. We agree here as well. This aspect of the project is scheduled for a subsequent and coming study.
Round 2
Reviewer 1 Report
Comments and Suggestions for Authors
no further comments
Reviewer 2 Report
Comments and Suggestions for Authors
Please, read very well the manuscript and do control of english